

# A strategy for the measurement of the $CO_2$ distribution in the stratosphere

Massimo Carlotti[1], Bianca Maria Dinelli[2], Giada Innocenti[1], Luca Palchetti[3]

[1]Dipartimento di Chimica Industriale "Toso Montanari", Universita' di Bologna, Italy
[2]Istituto di Scienze dell'Atmosfera e del Clima, ISAC-CNR, Bologna, Italy
[3]Istituto Nazionale di Ottica, INO-CNR, Sesto Fiorentino (FI), Italy

*Correspondence to*: Massimo Carlotti (Massimo.Carlotti@unibo)

**Abstract.** In this study we introduce a new strategy for the measurement of $CO_2$ distributions in the stratosphere. The proposed experiment is based on an orbiting limb sounder that measures the atmospheric emission within both the Thermal
InfraRed (TIR) and Far-InfraRed (FIR) regions. The idea is to exploit the contribution of the pure rotational transitions of molecular oxygen in the FIR to determine the atmospheric fields of temperature and pressure that are necessary to retrieve the distribution of $CO_2$ from its vibro-rotational transitions in the TIR. The instrument considered to test the new strategy is a Fourier transform spectrometer with two output ports hosting a FIR detector devoted to measure the $O_2$ transitions, and a TIR detector devoted to measure the $CO_2$ transitions. Instrumental and observational parameters of the proposed experiment
have been defined by exploiting the heritage of both previous studies and operational limb sounders. The performance of the experiment has been assessed with two-dimensional retrievals on simulated observations along a full orbit. For the purpose optimal spectral intervals have been defined using a validated selection algorithm. Both accuracy and spatial resolution of the obtained $CO_2$ distributions have been taken into account in the results-evaluation process. We show that the $O_2$ spectral features significantly contribute to the performance of $CO_2$ retrievals, and that the proposed experiment can determine two-
dimensional distributions of the $CO_2$ volume mixing ratio with precisions of the order of 1 ppmv in the 10-50 km altitude range. The error budget estimated for our test-case indicates that, in the 10-50 km altitude range, the accuracy of the $CO_2$ fields is set by the random component. The same accuracy can be obtained at higher altitudes provided the retrieval system is able to model the non-local-thermal-equilibrium conditions of the atmosphere. The best performance is obtained at altitudes between 20 and 50 km where the vertical resolution ranges from 3 km to 5 km and the horizontal resolution is of the order of
300-350 km depending on latitude.

## 1 Introduction

The role of $CO_2$ in the radiative budget of the biosphere is well-established as well as the consequences of its growth on the Earth's climate. The knowledge of $CO_2$ amounts and distribution is then essential to monitor the evolution of global warming and to drive the appropriate political choices. The remote sensing techniques with instruments on board of orbiting
platforms have demonstrated to represent a powerful strategy for monitoring the atmospheric state. In the case of $CO_2$, it can





be easily detected in the infrared spectral region (both Near InfraRed, NIR and Thermal InfraRed, TIR), where the spectral features generated by its vibro-rotational transitions are well visible. A number of space missions have been designed to measure $CO_2$ from space (Chedein et al., 2003, Crevoisier et al., 2004, Crevoisier et al., 2009, Kulawik et al., 2010, Buchwitz et al., 2005, Kuze et al., Boesch et al., 2011); most of them aim at measuring tropospheric abundances and use the

5 nadir observation geometry that is suitable for column measurements. Limb observations are more appropriate than nadir observations to discriminate the atmospheric behavior at different altitudes. Actually, limb-scanning spectrometers have been widely used to obtain the distribution of physical and chemical parameters in the stratosphere and upper troposphere, and to monitor their evolution over the years. Some limb experiments (e.g. the Michelson Interferometer for Passive Atmospheric Sounding (MIPAS), Fischer et al., 2008 and the High Resolution Dynamic Limb Sounder (HIRDLS), Gille et

10 al., 2008) exploit the $CO_2$ vibro-rotational transitions to determine pressure (P) and temperature (T) fields that, besides their intrinsic importance, are necessary in the data analysis to retrieve any target molecule from its spectral features. In these experiments the P, T retrieval process is based on the assumption that the $CO_2$ distribution is known; assumption that prevents the $CO_2$ spectral features to be used for the retrieval of its own atmospheric distributions. To date, only the Atmospheric Chemistry Experiment (ACE - Bernath et al., 2005) has provided daytime measurements of the $CO_2$ altitude

15 distribution between 5 and 25 km and in the mesosphere (Foucher et al., 2011, Sioris et al., 2014). In these measurements the accuracy of a single profile is of the order of 10 ppmv (Foucher et al., 2009). The big challenge for ACE (that performs occultation measurements in the TIR region) was to retrieve P and T (and pointing information at low altitudes) from spectral signals that are generated by sources different from the $CO_2$ transitions. A thorough discussion about the strategies that space experiments have used to measure $CO_2$ in the atmosphere, can be found in Bernath et al., 2005.

20 In this study we propose a new strategy for the measurement of $CO_2$ atmospheric distribution using a passive orbiting limb sounder. The starting idea is to exploit the pure rotational transitions of molecular oxygen ($O_2$) in the Far-InfraRed (FIR) region for the retrieval of the P and T fields that are necessary to retrieve the $CO_2$ distribution from its transitions in the TIR region. This strategy is expected to relax the strong correlation between T and $CO_2$ volume mixing ratio (VMR) that is intrinsic in the analysis of the TIR spectral features of $CO_2$. As the $O_2$ rotational transitions originate from a magnetic dipole

25 moment, their line strength is very low; nevertheless, due to both the huge abundance of $O_2$ and the long optical path of limb observation geometries, they are among the most prominent features of the atmospheric spectrum below 200 cm$^{-1}$. This is illustrated in Fig. 1 where the upper panel shows the simulated FIR atmospheric emission at 20 km tangent altitude (as observed by the spectrometer described in Sect. 2) and the lower panel shows the FIR emission when only $O_2$ is included in the simulation. Therefore we suggest to use an instrument capable to measure simultaneously the FIR atmospheric $O_2$ lines

30 and the TIR $CO_2$ lines to determine the VMR of $CO_2$ stratospheric fields with the target accuracy of 1 ppmv. For the purpose we propose to exploit a Fourier Transform (FT) spectrometer with two output ports respectively hosting a FIR detector to measure $O_2$ pure rotational transitions, and a TIR detector to measure $CO_2$ vibro-rotational transitions in the 700-900 cm$^{-1}$ region (roughly corresponding to MIPAS band A). The preliminary question to answer in our study is about the real contribution that the $O_2$ transitions can provide to the precision of the retrieved $CO_2$ VMRs. Actually, at stratospheric



temperatures, the Planck function and its sensitivity to T are much lower in the FIR region than in the TIR region where $CO_2$ features will have to be analyzed (the steep growth of the Planck function can be appreciated in the upper panel of Fig. 1). This suggests that the precision of the T values retrieved from only FIR transitions could not be sufficient to satisfactorily determine the $CO_2$ distribution from its TIR transitions. Moreover, a crucial step is to define experimental and retrieval

conditions that are suitable to determine $CO_2$ VMRs with the required precision in the stratosphere and in the upper troposphere (extending as low as possible the altitude range at which these precisions are achievable).

In Sect. 2 of this paper we define the instrumental and observational parameters of the proposed experiment (that we will denote as OXYCO2) and we discuss the selection of the optimal spectral intervals to be used for the analysis on simulated observations. Section 3 is devoted to describe the strategy adopted for the retrievals on simulated observations while the

results of the retrieval tests are shown and discussed in Sect. 4. Finally, in Sect. 5 we draw the conclusions of our study.

## 2 Operational parameters

### 2.1 Instrumental and observational parameters

The instrumental and observational parameters for OXYCO2 have been defined by exploiting the heritage of previous experiments, whether implemented (MIPAS, Fischer et al., 2008, and SAFIRE (Spectroscopy of the Atmosphere by Far-

InfraRed Emission), Bianchini et al., 2004) or designed for future missions with the support of documented studies (PIRAMHYD (Passive Infra-Red Atmospheric Measurements of Hydroxil), 1998 and IRLS (Infrared Limb Sounder) of the PREMIER (Process Exploration through Measurements of Infrared and millimetre-wave Emitted Radiation) experiment, Kraft et al., 2011). The OXYCO2 experiment is based on a limb-scanning FT orbiting spectrometer with two output ports respectively hosting a FIR detector (to measure the $O_2$ emission), and a TIR detector (to measure the $CO_2$ emission). The

SAFIRE spectral resolution of 0.004 cm$^{-1}$ (FT) has been adopted for OXYCO2. This relatively high value is meant to avoid, as much as possible, systematic errors due to transitions of interfering species (see Sect. 4.4), and to enhance the sensitivity of the measurements. A similar spectral resolution was also suggested for the FT spectrometer in the PIRAMHYD study aimed at the measurement of the hydroxyl radical (OH) in the atmosphere.

The space platform is assumed in the same orbit of ESA/ENVISAT (ENVironmental SATellite) that is a sun-synchronous

nearly-polar orbit at an altitude of about 700 km; the orbit period is of about 101 min. The spectrometer measures the limb emission backward looking along the orbit track (as MIPAS and IRLS); this choice enables to use the GMTR (Geofit Multi Target Retrieval) two-dimensional (2-D) algorithm (Carlotti et al., 2006) in the data analysis stage (see Sect. 3). For OXYCO2 we envisage the use of one-dimensional (1-D) vertical-array detectors that allow the measurement of a whole limb-scan with a single stroke of the moving mirror (2-D array detectors were proposed for the IRLS spectrometer that,

moreover, was designed to measure out of the orbit plane). A further advantage of array detectors is that the pointing accuracy translates into an angular bias which is common to all the limb geometries and does not affect the altitude sampling step (i.e. the separation between adjacent tangent points). We assume the time of 15 s for each interferogram recording cycle;





it follows that in a full orbit we can collect about 400 limb-scans roughly separated by 100 km. The Field Of View (FOV) of about 3 km at tangent point, and the limb geometries are the same as MIPAS Full Resolution (FR) nominal mode (Fischer et al., 2008); tangent altitudes are separated by 3 km from 6 to 42 km and then 47, 52, 60, and 68 km). Finally, the Noise Equivalent Spectral Radiance (NESR) is assumed to be 5 nW/(cm$^2$ sr cm$^{-1}$) (consistent with the values proposed for IRLS

and PIRAMHYD). The values of the spectral resolution, interferogram recording time, horizontal atmospheric sampling, and NESR are strictly interconnected. Actually the recording time, that defines the atmospheric sampling step, is proportional to the spectral resolution and is conditioned by the detectors integration time that, in turn, affects the NESR. The NESR value assumed for this study is feasible with the optical layout of SAFIRE having an optical throughput of 1.5 cm$^2$ sr, and using cooled detectors with "noise equivalent power" of about 10$^{-12}$ W/√Hz. For the estimation of the NESR, the specifications of

unstressed Ge:Ga detectors have been used. These detectors have already been deployed in space missions such as in the ESA Infrared Space Observatory (ISO) [Lim et al.,1998] and require low temperatures that can be reached with active system as Joule-Thompson cryocoolers [Rogalski, 2012]. The main instrumental and observational parameters of OXYCO2 are reported in table 1.

## 2.2 Selection of optimal spectral intervals

The retrieval targets need to be defined before the selection of an optimal set of spectral intervals (denoted as MicroWindows, MWs) to be analyzed in the inversion process. In our case, the spectral features of $O_2$ and $CO_2$ will have to be inverted to retrieve the atmospheric distributions of P, T and $CO_2$ VMR (which is the ultimate target). The visual inspection of observations simulated with different spectral resolutions and for different tangent altitudes, shows that the main interference to the $CO_2$ spectral features is due to the $O_3$ transitions. The high spectral resolution adopted for OXYCO2

reduces the $O_3$ interference to a level that does not require the inclusion of $O_3$ VMRs in the retrieval state vector (see Sect. 4.4). Nevertheless, the observations simulated in the FIR show that the interference of $H_2O$ transitions with the $O_2$ spectral features cannot be avoided by acting on the spectral resolution alone. The size of the systematic errors induced by the water variability in the inversion process is such that we decided to include $H_2O$ VMRs in the state vector of our retrieval tests (see Sect. 4.1). For this reason the MWs selection system was operated for the retrieval of P, T, and $H_2O$ VMR from FIR

transitions.

The MWs used for the retrievals described in this paper were selected adapting to our specific needs the algorithm described in Dudhia et al., 2002. This algorithm requires the definition of the uncertainties associated with an a-priori knowledge of the state vector. The MW growing process starts from a "seed" consisting of a 2-D set of spectral points (the two dimensions being frequency and retrieval altitude) that (when inverted) carry an error budget with respect to the retrieval targets. Any

improvement of these errors with respect to the a-priori uncertainties is quantified by the "information gain" scalar (Dudhia et al., 2002). The initial set of spectral points is left to expand in both dimensions and the information gain is calculated for the extended set (including the additional spectral point or retrieval altitude). The new set is retained in the MW if this corresponds to an increase of information. The MW growth process continues until either the inclusion of additional





elements does not correspond to increase of information or the maximum allowed MW size has been reached (we allowed for a maximum of 0.5 cm$^{-1}$ that correspond to 125 spectral points). The next MW will grow around a new seed and its information gain will be evaluated with respect to the error budget of the previously selected MW. The search for new MWs continues until the information gain obtained introducing the new MW assumes values below a fixed threshold. In this

process, seeds are identified by scanning the full spectral band in search of the 2-D set that carries the highest information gain. The scanning procedure is repeated, in search of a new seed, after each new MW has been generated. A consequence of this MWs selection strategy is that different combinations of MWs may be used at each altitude of the vertical retrieval grid. We remark that the MWs selection process evaluates the error budget (random and systematic) by simulating retrievals with an algorithm that operates on individual limb-scans (1-D algorithm), whereas the performance of OXYCO2 will be

evaluated using the 2-D GMTR algorithm (see Sect.s 3, 4 and appendix A) that operates on the whole set of limb-scans of the orbit.

Within the FIR measurements, 14 MWs were generated for the retrieval of P, T, and $H_2O$ VMR while 15 MWs were generated in the TIR for the retrieval of the $CO_2$ VMR. In order to limit the number of observations to be analyzed, a further selection was performed, among the initial set of 29 MWs, using the Information Load (IL) analysis (Carlotti et al., 2009,

Carlotti et al., 2013). The basic concepts of the IL analysis are summarized in Appendix A. Both intensity and altitude coverage of the IL (with respect to the target quantities) were the criteria used to evaluate the performance of each MW. In this way, the final set of observations used in the retrieval tests reported in Sect. 4 was reduced to 15 MWs: 6 of them in the FIR and 9 in the TIR. The frequency intervals of these MWs are reported in table 2 together with the corresponding number of spectral points and the altitude interval where the MW is used. Figure 2 shows maps of the overall IL of the selected sets

of MWs along a whole orbit. The IL values in Fig. 2 (as well as all the tests reported in this paper) have been calculated for the April 2004 atmosphere taken from the IG2 climatological database (Remedios et al., 2007). In all the maps of this paper (Fig.s 2, 4, and 8) the ground projection of the orbit track is represented by the "Orbital Coordinate" (OC): this is the angular polar coordinate originating at North Pole (OC=0 and 360 deg) and spanning the whole orbit plane (the South Pole has OC=180 deg while the Equator has OC=90 and 270 deg at the descending and ascending nodes respectively). In Fig. 2 the

upper-left panel shows the IL with respect to T of the 6 FIR MWs while the upper-right panel shows the IL with respect to T of the 9 TIR MWs. In the same figure, the lower-left panel shows the IL with respect to the $CO_2$ VMR of the TIR MWs. In order to interpret these results, the lower-right panel of Fig. 2 reports the T field of the atmospheric model used in our computations. The comparison of the maps of the IL with respect to T for the TIR and FIR MWs (note the factor of two between the scale expansions of the two upper panels of Fig. 2) highlights how larger the information about T is in the TIR

region. Furthermore, comparing the maps of the temperature IL of the TIR MWs with the T field we can appreciate the correlation between their minima. On the contrary, the temperature IL of the FIR MWs does not show clear correlations with the T field; this is a consequence of the T dependence of the Planck function ($P_f$). Actually, at the temperature of 250 K (average T in the stratosphere) $dP_f/dT$ is of the order of 25 nW/(cm$^2$ sr cm$^{-1}$ K) in the FIR while the numerical value of the



derivative rises to about 115 in the TIR (around 800 cm$^{-1}$). Finally, in Fig. 2 we can notice that the IL with respect to T of the whole set of analyzed observations is much higher than the IL with respect to the $CO_2$ VMR.

## 3 Layout of the retrievals on simulated observations

Retrieval tests have been carried out on simulated observations using the 2-D GMTR (Geofit Multi-Target Retrieval) analysis system (Carlotti et al., 2006) that was specifically developed for MIPAS measurements. The GMTR implements the Geo-fit algorithm (Carlotti et al., 2001a) in which the atmospheric state is allowed to vary in both the vertical and horizontal (latitudinal) dimension. With this approach the horizontal gradients are properly modeled by the simultaneous fit of measurements from a complete orbit. A further feature of GMTR is the Multi Target Retrieval (MTR) capability (Dinelli et al., 2004) that allows the simultaneous retrieval of atmospheric targets. The MTR strategy avoids the propagation of systematic errors induced by the overlapping of the spectral features of different retrieval targets when they are retrieved sequentially. We remark that when the MTR is applied to our case-study (see Sect. 4.1), the information on P and T is gathered from all the spectral features and not only from those of $O_2$ and/or $CO_2$. For the reader's convenience, Appendix A reports the basic physical and mathematical concepts behind the GMTR analysis system.

The steps of our retrievals on simulated observations are:

1.  Generate simulated observations with the GMTR standalone forward model; a random noise (with the instrumental round-mean-square deviation) is added to the synthetic spectra. "Reference" profiles are used at this stage to provide the atmospheric model.

2.  Retrieve geophysical targets from the simulated observations starting from an initial guess obtained by applying random perturbations to the reference profiles. Auxiliary data are the same used at point 1 so that their uncertainties are not taken into account in the systematic error budget.

3.  Assess the retrieval precision by comparing the retrieved values with the reference values used to generate the simulated observations. In our tests the retrieval precision includes the propagation of spectral noise and the effect of ray tracing. Actually, the 2-D ray tracing is different in the retrieval step with respect to step 1 because the perturbed values of P and T cause a different behavior of the refraction index and therefore slightly different paths of the lines of sight of the instrument.

4.  Assess the spatial resolution of the retrieval products through the 2-D averaging kernel of the inversion process.

In simulated retrievals a crucial issue is the strategy adopted (step 2) for the generation of the initial guess for the target quantities; that is the algorithm used for the perturbation of the reference profiles. A possible approach consists in applying a random percent perturbation (having pre-defined maximum amplitude) to each altitude of the reference profiles. This strategy often leads to unphysical atmospheric fields. This drawback is amplified in the 2-D approach where the initial guess of the retrieval is a set of adjacent vertical profiles and the random perturbations at each altitude may also produce unphysical horizontal gradients of the target quantities. On the other hand, if a constant perturbation is applied to the





elements of a whole profile, unphysical shapes are avoided in the vertical domain but still unphysical horizontal gradients may be present. For these reasons we have implemented the following perturbation strategy:

once defined the maximum amplitude of the perturbation (**B**), we apply a percent perturbation (*pert*) to each altitude (*h*) of the reference profiles given by the sinusoidal function:

$$pert(h) = A * \sin\left(2\pi \frac{h}{Z} + \phi\right) \quad (1)$$

where:

*A* is a random number, associated to each altitude profile, $\leq$ **B**

*Z* = 80 km,

$\phi$ is a random number, associated to each altitude profile, ranging from 0 to $2\pi$.

This recipe generates a sinusoid of random amplitude (*A*) and phase ($\phi$) that introduces a smooth variation (with altitude) of the perturbation. Since the reference (climatological) profiles are generally smooth, our perturbation strategy leads to physical shapes of the perturbed profiles. However, depending on the random value of $\phi$, a perturbed profile will have altitudes that are affected by the maximum perturbation (*A*) and others with null perturbation. In order to assess the behavior

of our simulated retrievals with respect to a balanced altitude distribution of the perturbations, we then need to perform a number of retrieval tests which is large enough to make the perturbation almost constant with altitude when averaging the results. Figure 3 shows average altitude distributions of *pert*(*h*) (generated with Eq. (1) ) after a set of 20 GMTR simulated retrievals. The five black lines of Fig. 3 correspond to perturbations calculated for **B** = 1, 2, 3, 4, 5 respectively (values that will be used in our tests described in Sect. 4). Due to the strong variability of the tropospheric $H_2O$ VMR, a value of **B** which

is constant at all altitudes is not appropriate to perturb the $H_2O$ VMR profiles. Therefore for this target we have used **B** = 30 at the lowest altitude; this value is linearly decreased up to 15 km where it becomes 10 and remains constant above. The red line in Fig. 3 (measured by the upper scale) refers to the average perturbations applied to $H_2O$ VMR profiles. In Fig. 3 we see that the average over 20 GMTR runs generates a satisfactory approximation of the constant percent values of 0.3, 0.7, 1.0, 1.3, 1.5 that correspond to **B** = 1, 2, 3, 4, 5 respectively. The results of 20 runs have been averaged in all the retrieval

tests reported in Sect. 4.3.

## 4 Retrieval tests

### 4.1 Retrieval strategy

Preliminary tests were carried out with the "sequential retrievals" strategy previously used in the MIPAS data analysis system (Ridolfi et al., 2000). In the case of MIPAS, T and P (at tangent points) are first retrieved from the $CO_2$ spectral

features; the VMR of atmospheric species is then derived in subsequent steps by using the previously retrieved parameters as known quantities in the forward model. In our tests P, T and $H_2O$ VMR were first derived from the FIR MWs and then used



as known quantities in the retrieval of the $CO_2$ VMR from its TIR spectral features. This strategy led to $CO_2$ VMR precisions that did not approach the target value. We have verified that the insufficient precision of the retrieved T fields is responsible for the bad performance of the "sequential" strategy; the evidence is given by the precision of the $CO_2$ VMRs that becomes satisfactory when, in the $CO_2$ analyses, we use the retrieved values of P and $H_2O$ VMR but the reference values of T.

The retrieval strategy was then changed to fully exploit the potentiality of both the measurement strategy and the MTR functionality of the retrieval system: FIR and TIR observations were simultaneously analyzed to retrieve in a single step P, T, and the VMRs of $H_2O$ and $CO_2$. As mentioned in Sect. 3, the advantage of MTR is that the information on P and T is gathered from all the analyzed spectral features (including $O_2$, $CO_2$ and $H_2O$). With this approach, the T dependence of the $O_2$ transitions relax the correlation between T and $CO_2$ VMR that is intrinsic in the analysis of $CO_2$ spectral features alone. A

second advantage comes from the fact that, in our retrieval setup, the dominant information about T comes from the shape of the Planck function rather than from the dependence of the line strengths from T (Carlotti et al., 2013). If we look at the FIR and TIR spectra as two independent measurements sampling the Planck function in separated frequency regions, the retrieval of T results better constrained when simultaneously analyzing both regions. On the other hand, the drawback of the MTR strategy is in the additional set of retrieval parameters introduced in the state vector by $H_2O$ VMRs and atmospheric

continuum parameters associated to the FIR MWs (see Sect. 4.2).

## 4.2 Retrieval setup

In all the retrieval tests performed with the above mentioned strategy, the state vector contains the parameters describing the atmospheric field of the geophysical targets including the atmospheric continuum (below 27 km) at the frequency of each MW. The whole set of 15 MWs (see table 1) was used in all tests reported in the next sub-section. By considering that not all

the MWs are used at each of the 17 tangent altitudes, a total of 18,904 spectral points are analyzed for each limb-scan; this corresponds to a total of 7,580,504 spectral points simultaneously fitted in the GMTR analysis of a full orbit. Our retrievals were carried out on a vertical grid coincident with the tangent altitudes of the limb-scans except for the 9 km altitude that was removed in order to improve the precision in the upper troposphere; this exclusion degrades the vertical resolution at tropospheric altitudes (see Sect. 4.3). The horizontal grid was defined with altitude profiles separated by 2 latitudinal

degrees; this choice implies 180 profiles in the whole orbit that corresponds to a horizontal retrieval sampling which is twice wider than the measurement sampling. This setup leads to 2880 retrieval parameters for each of the four main targets. The number of continuum parameters is 13320 (this number does not match the product between number of retrieval altitudes, number of MWs, and number of limb-scans because not the all MWs are used at all altitudes). The total number of retrieval parameters is then 24840.

The GMTR analysis system makes use of the maximum a-posteriori likelihood technique (optimal estimation). Since the initial guess of the retrieval parameters was obtained by applying "strong" perturbations (see Sect. 3), the reference profiles were used as a-priori estimate of the state vector. However, large uncorrelated uncertainties (diagonal elements of the a-priori VCM matrix) were associated to the a-priori values; they are 30% for P, a constant error of 7 K for T, 80% for the



VMR, and 100% for atmospheric continuum parameters. A constant term was also added to the $H_2O$ and atmospheric continuum a-priori errors to prevent strong constraints where the a-priori fields assume very small values. Dedicated tests have shown that the retrieval results barely depend from the errors assigned to the a-priori estimates.

### 4.3 Retrieval results

The IL analysis reported in Sect. 2.2 shows that the analyzed observations are far more sensitive to T than to the $CO_2$ VMR. Therefore, in order to assess the stability of the retrieval system and its capability to sort out the information between T and $CO_2$ VMR in unfavorable conditions, we have also considered strongly perturbed atmospheric fields as initial guess for our retrieval tests (even if the perturbations might appear unrealistic). The initial guess atmospheric fields were generated by applying to the reference profiles the perturbations generated by Eq. (1) where values of $B$ = 2, 3, 4, 5 were used for $CO_2$

VMR profiles while different combinations of $B$ values were used for the perturbation of P and T profiles. The impact of the perturbation adopted for the $H_2O$ profiles (see Sect. 3) has been evaluated by performing retrievals keeping fixed the perturbation of P, T and $CO_2$ but using different magnitudes of $B$ for the $H_2O$ perturbations: the difference between the retrieved $CO_2$ fields resulted to be negligible, therefore the perturbation reported in Fig. 3 for $H_2O$ was kept constant in all our tests.

In order to provide a typical example of the retrieval results, Fig. 4 shows the 2-D distribution of the average difference (absolute value) between retrieved and reference $CO_2$ VMRs in the case of P, T, and $CO_2$ initial guess profiles obtained with $B$ = 2, 1, 4 respectively (this example will be referred to as "sample test" in the following part of the paper). An overall picture representing the results shown in Fig. 4 can be given in the form of a plot (that we call *overall plot*) reporting, at each altitude, the average value of the differences in Fig. 4 along the whole orbit. Retrieval results in term of *overall plots* will be

shown for the case of "weak" and "severe" perturbations of P and T profiles. The results obtained in the case of "weak" P, T perturbations ($B$ = 2 for P and $B$ = 1 for T) are reported in Fig. 5. In the four panels of this figure the red lines shows the *overall plot* for the $CO_2$ retrievals obtained by perturbing its reference profiles with values of $B$ equal to 2 (top left), 3 (top right), 4 (bottom left), and 5 (bottom right). Figure 6 reports the same quantities of Fig. 5 but in the case of "severe" P, T perturbations ($B$ = 3 for P and $B$ = 4 for T). The red curves in Fig.s 5 and 6 suggest that the retrieval is rather stable with

respect to the size of the applied perturbations and that the $CO_2$ VMRs can be retrieved by the OXYCO2 experiment with precisions that are of the order of 1 ppmv between 10 and 60 km. To fully understand the importance of measuring FIR and TIR spectral regions together, we need to assess whether and to what extent the FIR observations are necessary and contribute to the precision obtained in our retrieval tests. In order to answer this question we have excluded the FIR MWs from the set of the analyzed observations. Since the $H_2O$ contribution is negligible in the TIR MWs, we have also excluded

the $H_2O$ parameters (and the atmospheric continuum of FIR MWs) from the state vector of the retrieval. In these conditions the total number of observations analyzed along the full orbit is reduced to 2,663,041 while the total number of retrieval parameters is 15,660. The green lines in Fig.s 5 and 6 report the *overall plots* of this modified retrieval setup in the case of "weak" and "severe" P, T perturbations ($H_2O$ profiles are unperturbed in this case). The results obtained without the FIR





measurements indicate that the contribution of FIR observations to the retrieval precision is important and increases by increasing the size of the $CO_2$ perturbations.

Another important issue is the spatial resolution of the retrieval products of the OXYCO2 experiment. The upper panels of Fig. 7 report, for the "sample test", the vertical (left) and horizontal (right) resolution of the retrieved $CO_2$ VMR field; they

are calculated using the 2-D averaging kernel matrix as described in Carlotti et al., (2007). Since the correlation between $CO_2$ VMR and T is a major issue in our study, the same quantities as in the upper panels are reported in the lower panels of Fig. 7 for the retrieved T field. It can be seen in Fig. 7 that the vertical resolution of the retrieved $CO_2$ field is consistent with the geometrical sampling of the atmosphere (3 km) between 20 and 50 km. The horizontal resolution mostly exceeds the geometrical sampling of the retrieval grid with values lower than 350 km in the 20 - 50 km altitude range (to be compared

with the 220 km of the retrieval grid). The complex interdependence between the many variables of the MTR inversion makes difficult the interpretation of the 2-D distributions in the maps of Fig. 7. However, by looking at the IL maps of Fig. 2 and at the maps of Fig. 7 we can identify some correlation between the distributions of information load and those of spatial resolution.

## 4.4 Error budget

In this sub-section we provide an estimate of the systematic error components that are associated to the choices implemented in the OXICO2 experiment. As stated in Sect. 2.2, the MWs selection algorithm provides an estimate of the total error budget together with the contribution of the individual sources. This budget indicates that the dominant components of the systematic error are the uncertainties on the VMRs of $O_3$, $NO_2$ and $HNO_3$, and the approximation introduced by the Non Local Thermal Equilibrium (NLTE) which is not modeled by the retrieval system. Since the 1-D retrieval algorithm used for

the MWs selection differ in several aspects from GMTR, we have estimated the systematic errors introduced by the interfering species by repeating the "sample test" with the VMR profiles of $O_3$, $NO_2$ and $HNO_3$ perturbed according to their climatological variability. In the case of Non Local Thermal Equilibrium (NLTE) the "sample test" was carried out on observations simulated by using the vibrational temperature profiles of the target species along the full orbit (that is irrespective of day-nighttime conditions). We remark that, since we do not impose hydrostatic equilibrium in our retrievals, a

possible constant pointing bias (due to the use of 1-D array detector (see Sect. 2.1)) translates into the retrieval of an effective value of P at the vertical retrieval grid points. Therefore, if the resulting $CO_2$ VMR values are represented as a function of P, any bias in the pointing will be corrected. On the other hand, the errors on P have a negligible impact on the retrieved $CO_2$ VMRs (see Sect 4.1). About the errors induced by the uncertainty of spectroscopic parameters (in particular those of $CO_2$ and $O_2$) we notice that, if such an experiment will be considered for operational implementation, there will be

dedicated measurements of them.

Figure 8 reports the main components and the total systematic error associated to the full set of 15 MWs used in our retrieval tests. The red curves of Fig.s 5 and 6 compared to the total systematic error of Fig. 8 show that, below 50 km, the contribution of the systematic component of the total error is negligible with respect to the random component. However, the



interference of the $O_3$ transitions turns out to be the most important source of systematic error below 40 km. This outcome justifies the relatively high spectral resolution adopted for OXYCO2 (see Sect. 2.1). Figure 8 also shows that above 50 km the capability of modeling NLTE is a limiting condition for the retrieval accuracy.

## 5 Conclusions

In this paper we have studied the possibility to measure the $CO_2$ distributions in the stratosphere by exploiting the synergism between FIR and TIR limb-sounding measurements from space. The pure rotational transitions of $O_2$ in the FIR region are exploited to relax the strong correlation between T and $CO_2$ VMR that is intrinsic in the analysis of $CO_2$ spectral features in the TIR region. In order to assess the performance of this concept, we have considered a hypothetical experiment (denoted as OXYCO2) based on a FT spectrometer for which we have defined instrumental and observational parameters. The proposed

experiment is inspired by both operational experiments and previous studies for space limb sounders. The ability to measure $CO_2$ distributions was established through retrievals on OXYCO2 observations simulated on optimal spectral intervals.

We have shown that accurate T fields are necessary to retrieve $CO_2$ VMRs from the vibro-rotational spectra of this molecule with precisions of the order of 1 ppmv. For the analysis of FIR observations, the VMR of $H_2O$ must be included in the state vector in order to avoid the systematic errors due to the interference of its spectral features. The best performance of the

simulated retrievals is obtained when FIR and TIR observations are simultaneously analyzed to derive $CO_2$ fields together with P, T and $H_2O$ fields.

A number of retrieval tests were carried out with the 2-D GMTR algorithm starting from initial guess profiles, of the retrieval targets, obtained with different perturbations of the reference profiles. The outcome of these tests shows that a precision of the order of 1 ppmv can be achieved for the $CO_2$ VMRs in the altitude range between 10 and 60 km. These

results have been obtained, along a full orbit, onto a vertical grid defined at steps of 3 km up to 42 km and 5 km above, and onto a horizontal grid of profiles separated by about 220 km. The vertical resolution of the retrieved $CO_2$ fields is consistent with the retrieval grid between 20 and 50 km. The horizontal resolution depends on altitude and latitude and mostly exceeds the size of the retrieval grid; the best performance is in the 20 - 50 km altitude range where the horizontal resolution does not exceed 350 km.

If the FIR observations are excluded from the retrieval process, the precision of the $CO_2$ fields worsen by a factor which depends on altitude but may exceed 2 in the 20 - 40 km altitude range. This outcome demonstrates the starting hypothesis that FIR observations contribute to the $CO_2$ retrieval by introducing the information on T contained in the pure rotational spectrum of $O_2$.

The OXYCO2 experiment can determine two-dimensional distributions of the $CO_2$ VMR with precisions of the order of 1

ppmv in the 10-50 km altitude range. The assessment of the systematic errors associated to the set of analyzed MWs shows that below 50 km their contribution to the total error is negligible with respect to the random component. However, below 40 km the interference of $O_3$ transitions tends to be a meaningful source of systematic errors; therefore the spectral resolution



needs to be high enough to ensure the retrieval accuracy to be driven by the random component of the error. Above 50 km, the accuracy of the $CO_2$ fields mainly depends on the capability of the retrieval system to model NLTE conditions. We can conclude that $CO_2$ VMR profiles can be retrieved by the OXYCO2 experiment with accuracies that are of the order of 1 ppmv between 10 and 50 km of altitude.

Finally, we remark that the target of this study was the retrieval of $CO_2$ VMR fields. However, a number of side-products can be identified within the spectral ranges of the OXYCO2 experiment; first of all the hydroxyl radical (OH, of fundamental importance in atmospheric chemistry) that displays unique spectral features around 118 cm$^{-1}$ (Carlotti et al., 2001b). For the analysis of these features the FIR spectroscopy was identified as the best measurement strategy (PIRAMHYD, 1998).

**Appendix A**

**The GMTR analysis system**

The GMTR analysis system (Carlotti et al., 2006) was developed as an open source code specifically designed for MIPAS measurements. It is based on the Geo-fit approach (Carlotti et al., 2001a) upgraded with the Multi-Target Retrieval (MTR) functionality (Dinelli et al., 2004). With the Geo-fit all the observations collected along the orbit track are analyzed

simultaneously by exploiting the continuous repetition of the limb sequences along the orbit plane. This repetition allows the gathering of information about a given parcel of atmosphere from all the observations whose line of sight crosses that parcel whatever limb-sequence they belong to. This concept is illustrated in Fig. A1. Since the loop of overlap between nearby sequences closes when the atmospheric parcel sounded by the first sequence (preferably at North or South Pole) is observed again at the end of the orbit, the full gathering of information is achieved by merging in a simultaneous analysis the

observations of the complete orbit. In the Geo-fit the discretization of the atmosphere is operated on both the vertical and the horizontal dimensions (Carlotti et al., 2001). In the vertical dimension altitude levels delimit atmospheric layers as in a 1D approach. The horizontal discretization is built using segments (denoted as "radii") perpendicular to the Earth's geoid and extended up to the boundary of the atmosphere. The 2D discretization leads to a web-like picture in which consecutive levels and radii define plane figures denoted as "cloves" while the points defined by the crossing of levels and radii are denoted as

"nodes" (Fig. A1 depicts the result). The value assumed by physical and chemical atmospheric quantities is assigned to each node by interpolating the profiles of the adopted atmospheric model. Therefore the 2-D discretization enables to model horizontal atmospheric structures. The retrieval grid of the Geo-fit is independent of the measurement grid so that atmospheric profiles can be retrieved with horizontal separations different from those of the measured limb-scans. This strategy makes it possible to select the optimal horizontal resolution of the retrieved profiles on the basis of its trade-off with

the precision of the retrieval parameters (Carlotti et al., 2007).





In the GMTR algorithm the observations are analyzed with a constrained non-linear least squares fit based on the Gauss-Newton method (Ridolfi et al., 2000, Carlotti et al., 2006, Carlotti et al., 2007). The iterative solution expression of this method is given by:

$$x_{i+1} = x_i + (K_i^T S_n^{-1} K_i + S_a^{-1})^{-1}[(K_i^T S_n^{-1} n_i - S_a^{-1}(x_i - x_a)] \qquad \text{(A1)}$$

where:

- $i$ denotes the iteration index,

- $x_i$ is the state vector at iteration $i$,

- $x_a$ is the vector containing the a-priori state vector

- $n_i$ is the vector containing the differences between each observation and its value simulated by the forward model,

- $S_n$ is the variance-covariance matrix (VCM) of the vector $n$,

- $K_i$ is the matrix (Jacobian matrix) whose rows contain the derivatives of each observation with respect to all the retrieved parameters,

- $S_a$ is the variance-covariance matrix of the a-priori state vector,

At the last iteration the errors associated with the solution of the inversion procedure are characterized by the VCM of $x$ given by:

$$V_x = (K^T S_n^{-1} K + S_a^{-1})^{-1} \qquad \text{(A2)}$$

Matrix $V_x$ maps the measurement random errors (represented by $S_n$) onto the uncertainty of the values of the retrieved parameters; in particular, the square root of the diagonal element $p$ of $V_x$ provides the Estimated Standard Deviation (ESD) of the corresponding parameter:

$$ESD_p = \left[(V_x)_{p,p}\right]^{1/2} \qquad \text{(A3)}$$

**The information load analysis**

In the 2-D approach it is possible to associate to each clove of the atmospheric discretization a quantifier that measures the amount of information carried by that clove with respect to a retrieval target (Carlotti and Magnani, 2009). The information load quantifier $(\Omega)$ of clove $h$ with respect to atmospheric parameter $q$ is defined as (Carlotti and Magnani, 2009):





$$\Omega\left(q,h\right)=\left[\sum_{i=1}^{l}\sum_{j=1}^{m}\sum_{k=1}^{n}\left(\frac{\partial Y_{ijk}}{\partial q^{h}}\right)^{2}\right]^{1/2} \quad \text{(A4)}$$

where:

$Y_{ijk}$ = spectral radiance of observation geometry $i$ at wavenumber $k$ of the analyzed MW $j$,

$l$ = number of observation geometries that cross the clove $h$,

$m$ = number of spectral intervals analyzed in observation geometry $i$,

$n$ = number of spectral points in spectral interval $j$.

In Eq. (A4) the triple summation could be grouped in a single summation over all the spectral radiances that are affected by

the value of $q$ in clove $h$. Therefore the column vector containing the whole set of elements within the triple summation of

Eq. (A4) is the Jacobian matrix corresponding to the retrieval of the scalar value of $q$ in clove $h$ and (moving to the lower-

case notation for $K$ that represents now a vector) we can write Eq. (A4) as:

$$\Omega = (k^{T}k)^{1/2} \quad \text{(A5)}$$

On the other hand, if we assume an unconstrained retrieval on observations that are uncorrelated and characterized by

constant uncertainty $S_{n}= I$, Eq. A(2) becomes:

$$V_{q} = (k^{T}k)^{-1} \quad \text{(A6)}$$

Hence, it follows from Eq. (A3) that the ESD of the retrieved value of $q$ in clove $h$ is given by $1/\Omega$. These considerations

justify the quadratic summation as combination rule of the partial derivatives in Eq. (A4). Further on, if the observations are

correlated and characterized by the VCM $S_{n}$, $\Omega$ can be calculated as:

$$\Omega = (k^{T}S_{n}k)^{1/2} \quad \text{(A7)}$$

The values of the information load can be calculated for each clove of the 2D discretization so that we can draw, for each

retrieval target, a map of the 2D distribution of the $\Omega$ quantifier. The information load maps provide a picture of the "real"

atmospheric sampling of the observations; they can be used to define optimal retrieval grids (where the information peaks) or

to compare the atmospheric sampling for different targets or for different observation strategies. The information load

analysis is therefore suitable to study the performance of different experimental conditions.



*Acknowledgements*. The MWs selection task was carried out by exploiting an "Erasmus+ traineeship 2014/2015" grant, at Oxford University, with the supervision of Anu Dudhia that we thank for his valuable contribution.

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



**Table 1.** Instrumental and observational parameters of OXYCO2

| Orbit | sun-synchronous polar |
|---|---|
| Detectors | 1-D array |
| Spectral coverage FIR | 80 - 180 cm$^{-1}$ |
| Spectral coverage TIR | 685 - 930 cm$^{-1}$ |
| Spectral resolution (FT) | 0.004 cm$^{-1}$ |
| Interferogram recording time | 15 s |
| Vertical sampling | MIPAS FR nominal mode |
| Horizontal sampling | ~110 km |
| NESR | 5 nW/(cm$^2$ sr cm$^{-1}$) |
| FOV at tangent point | 3 km |





**Table 2**. Definition of the MWs selected for the retrieval tests.

| | FIR | | |
|---|---|---|---|
| *MW number* | *frequency interval (cm⁻¹)* | *N. of points* | *Altitude coverage (km)* |
| 1 | 117.840 – 118.336 | 125 | 6 - 68 |
| 2 | 119.524 – 120.020 | 125 | 6 - 68 |
| 3 | 150.280 – 150.776 | 125 | 6 - 68 |
| 4 | 161.128 – 161.624 | 125 | 6 - 68 |
| 5 | 163.264 – 163.728 | 125 | 12 - 60 |
| 6 | 175.760 – 176.256 | 125 | 6 - 68 |

| | TIR | | |
|---|---|---|---|
| *MW number* | *frequency interval (cm⁻¹)* | *N. of points* | *Altitude coverage (km)* |
| 7 | 700.884 – 701.108 | 57 | 15 - 68 |
| 8 | 702.112 – 702.304 | 49 | 15 - 68 |
| 9 | 758.760 – 758.924 | 42 | 6 - 68 |
| 10 | 760.252 – 760.484 | 59 | 6 - 68 |
| 11 | 810.904 – 811.076 | 44 | 18 - 60 |
| 12 | 818.664 – 818.856 | 49 | 6 - 68 |
| 13 | 820.116 – 820.312 | 50 | 12 - 68 |
| 14 | 821.756 – 821.984 | 58 | 15 - 68 |
| 15 | 918.532 – 918.716 | 47 | 24 - 68 |





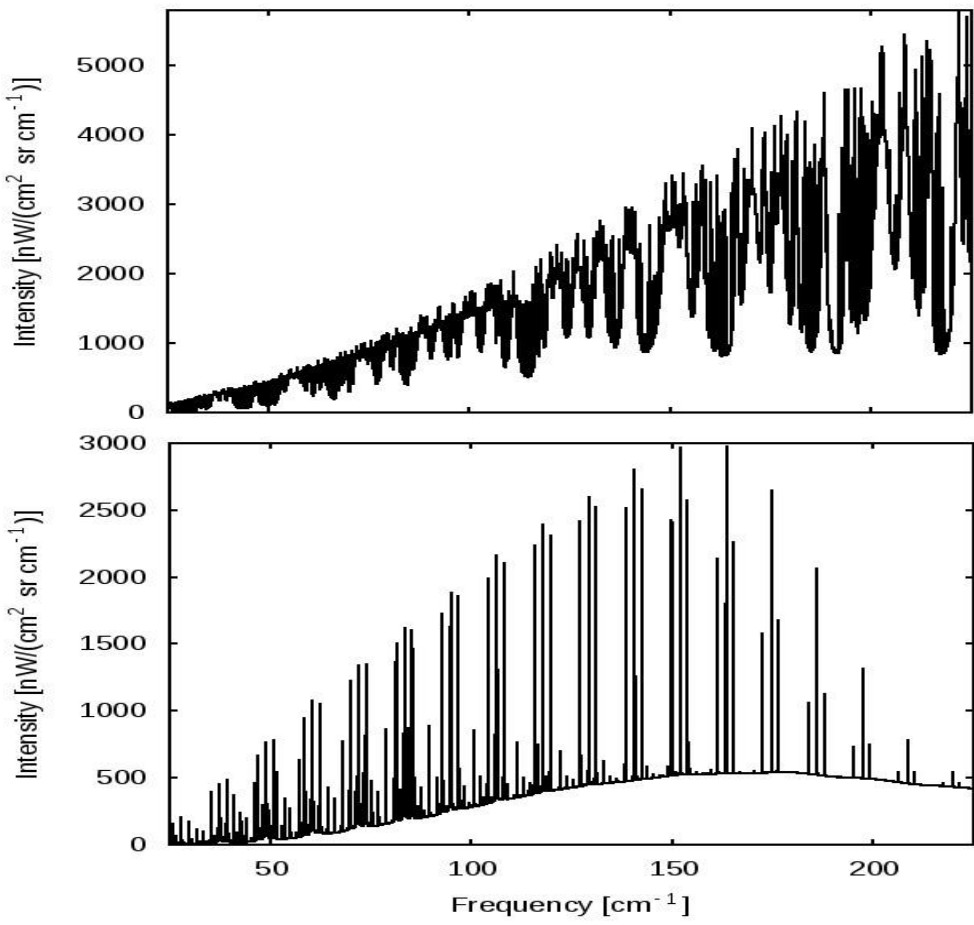

**Figure 1.** Upper panel: far infrared atmospheric spectrum simulated at 20 km tangent altitude, lower panel: same as upper panel but only $O_2$ is included in the simulation.





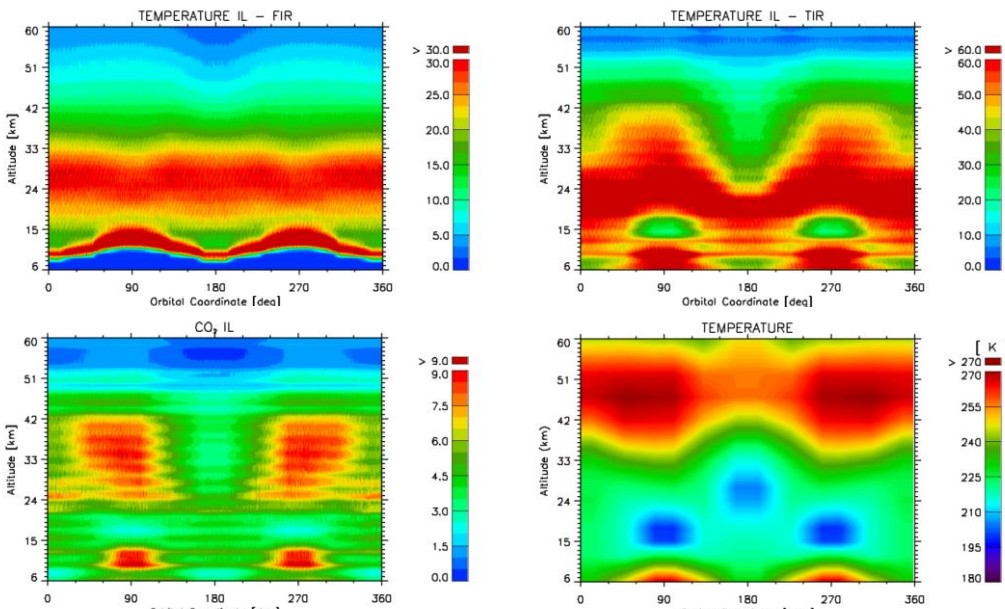

**Figure 2**. Upper left: IL with respect to T of the FIR MWs, upper right: IL with respect to T of the TIR MWs, lower left: IL with respect to $CO_2$ VMR of the TIR MWs, lower right: T field of the reference atmosphere.





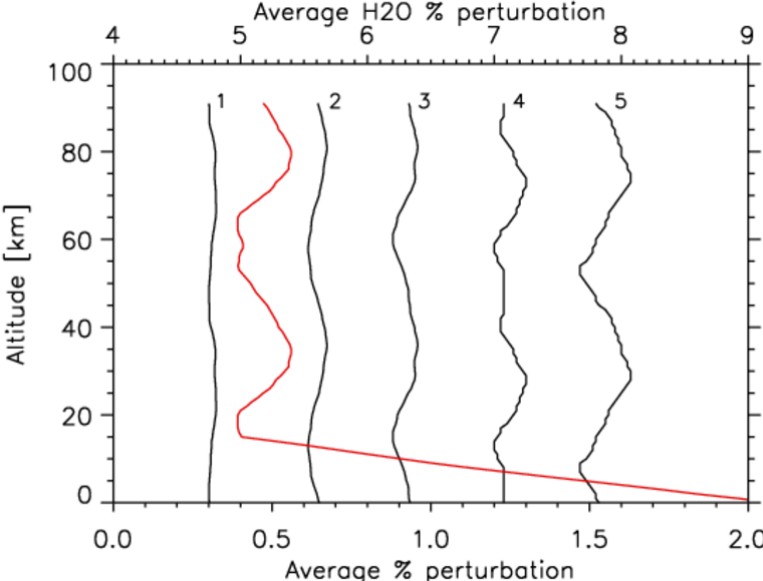

**Figure 3.** Average % values (over 20 GMTR runs) of $pert(h)$ corresponding to different values of $B$ in Eq. (1). The value of $B$ is reported on top of each line except for the red line that refers to $H_2O$ perturbations and is measured by the upper scale: in this case $B$ is 30 at 0 km and linearly decreases up to 15 km where it assumes the constant value of 10.




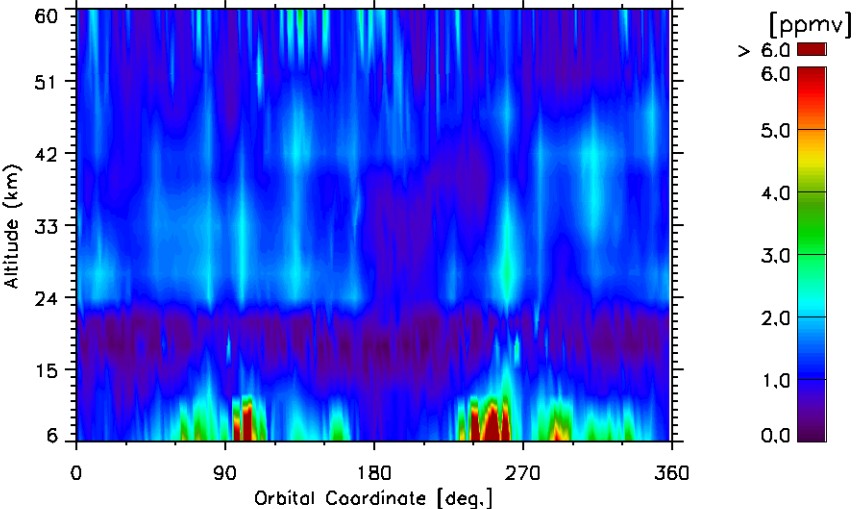

**Figure 4.** 2-D distribution of the averaged absolute difference between reference and retrieved $CO_2$ VMRs resulting from the "sample test" (see text).




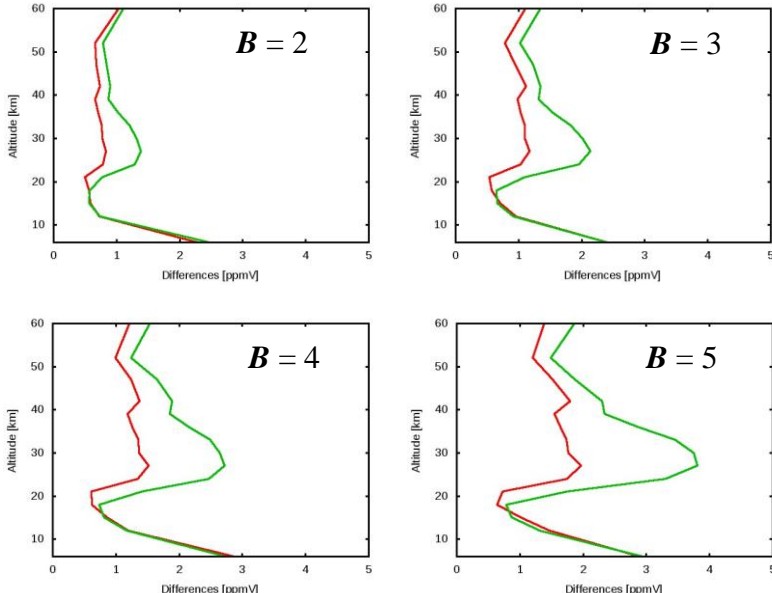

**Figure 5.** red lines: *overall plots* for $CO_2$ retrievals in the case of **$B$**=2 for P-, and **$B$**=1 for T-perturbations in Eq. (1). Each panel contains the **$B$** value used for the $CO_2$ VMR perturbations. Green lines: same as red lines but excluding FIR MWs and $H_2O$ parameters from the retrieval setup.





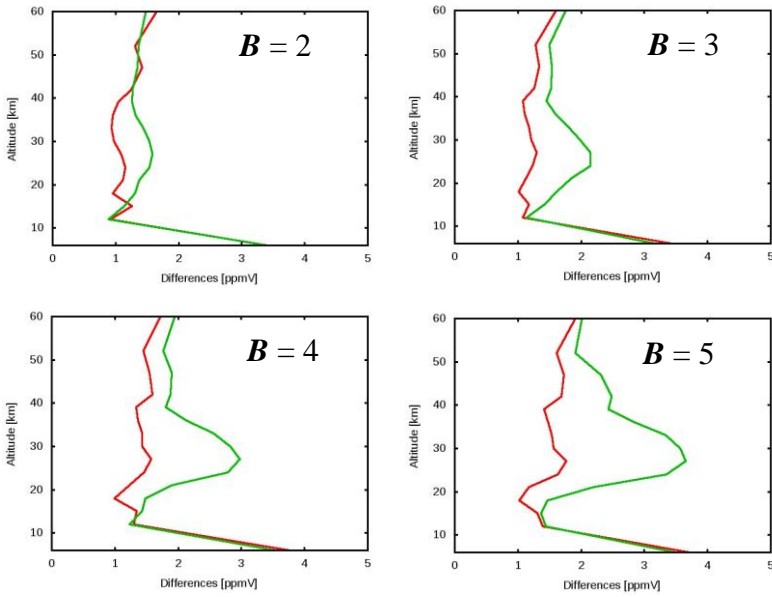

**Figure 6.** As Fig. 5 but in the case of $B = 3$ for the P- and B = 4 for the T- perturbations.





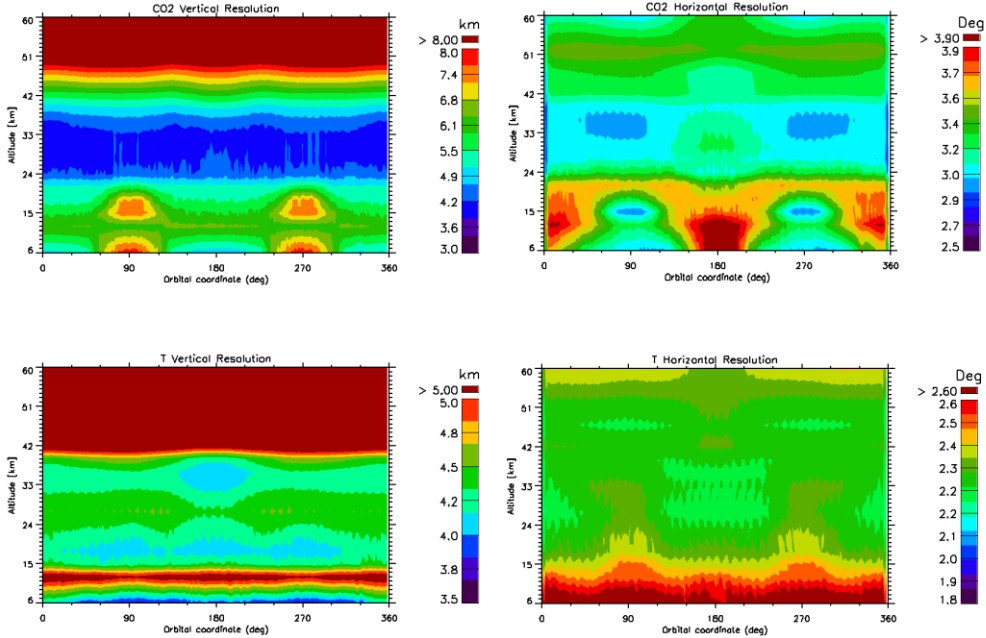

**Figure 7**. Maps of the vertical (left panels) and horizontal (right panels) resolutions as a function of altitude and orbital coordinate obtained with the averaging kernel matrix of the "sample test" retrieval. The upper panels refer to the retrieved $CO_2$ fields and the lower panels refer to T.





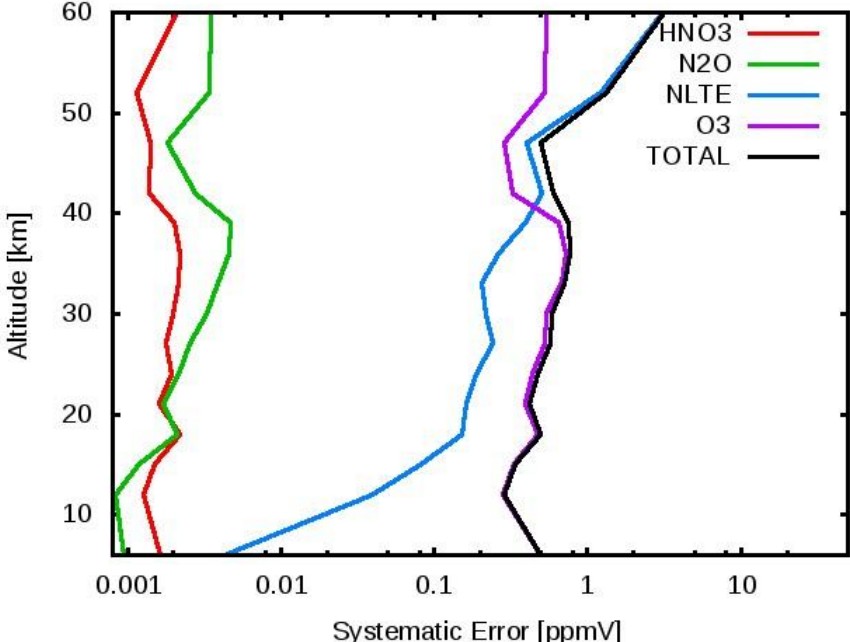

**Figure 8.** Error components and total systematic error associated to the set of 15 MWs that were used in retrieval tests. Only the three atmospheric gases whose abundance is the main source of error are reported. NLTE indicates the error propagated by neglecting non local thermal equilibrium conditions in the retrieval system.




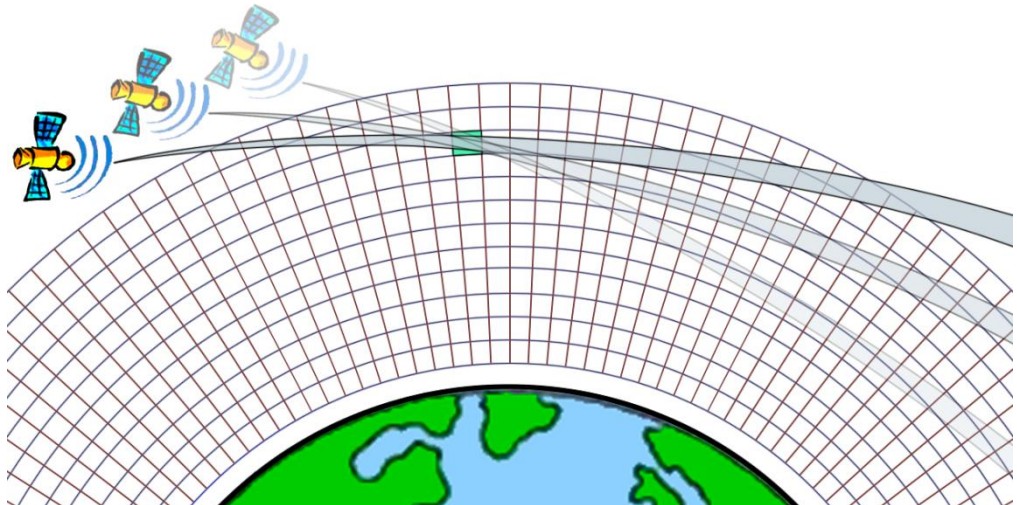

**Figure A1.** Picture illustrating the 2-D discretization of the atmosphere and the merging of information from different observation geometries in the Geo-fit analysis.