# Peer review of "A strategy for the measurement of the $CO_2$ distribution in the stratosphere"

_Atmospheric Measurement Techniques, 2016_

## Referee Comment (RC1) · Anonymous Referee #2 · 6 Aug 2016

This paper presents a reasonable strategy for retrieving the vertical distribution of CO2 in the stratosphere. An alternative strategy would be to use solar occultation instead of thermal emission for these same bands. This could improve the shot noise and the vertical resolution since:

1) The sun will always be a stronger source of radiation than the Earth's atmosphere at any wavelength.

2) For solar occultation, the vertical resolution is not tied to the vertical structure of the temperature (see Fig. 7 bottom left of Carlotti et al.), and therefore would not worsen severely in the tropopause region as is the case for thermal emission as expected.

For CO2, given the main interest of the authors to observe the long-term slight increases in VMR in the stratosphere, frequent measurements are not required and thus

space-based solar occultation could be applicable. The authors could consider the O2 lines in the TIR as well, which are useful in the stratosphere up to ∼38 km, based on ACE-FTS O2 retrieval accuracy. This could alleviate the need for two detectors regardless of whether thermal emission or solar occultation is used. There are three main problems with this paper:

1) The error budget is incomplete, specifically with regard to sources of systematic uncertainty.

2) O2 does not appear to be adding much p T information outside of the 20-35 km range, raising the question about strong correlations between CO2 VMR and T.

3) Related to problems 1 & 2, the authors present their method as one that has small systematic uncertainties and an accuracy of 1 ppm (P2L30 & P12L3), yet ignore many significant systematic sources of uncertainty and consider only minor ones.

This paper is publishable after major revisions. The major revisions involve accounting for more sources of systematic error and switching back to sequential retrievals (see below). Furthermore, the statement that CO2 VMR is retrievable to 1 ppmv between 10 and 50 km is grossly misleading in my opinion. There is no point to getting the community excited about an instrument that can supposedly measure CO2 profiles to ∼1 ppm, when it hinges on line intensities of CO2 to be measured to 0.25%. Also, it hinges on O2 line intensities to be measured extremely accurately: a 1% bias in O2 line intensities will lead to a 70 m bias in TH (or a 1% bias in pressure). This will translate to a ∼1% bias in CO2, much larger than the sources of systematic error that the authors have selected. The combined biases in CO2 and O2 spectroscopy could either cancel or lead to a 2% bias in the worst case.

The claim that dedicated spectroscopic measurements will be made in the future is not acceptable to me for the present manuscript. I would assume the spectroscopists previously involved in measuring line intensities were dedicated to achieving the best accuracies possible. See table 3 of Tashkun et al. (2015). Searching through the

systematic uncertainty column (2nd last column) of this table, I see values as low as 2% (e.g. by Delière et al., 2012) for the lines in a region overlapping the spectral region proposed by the authors. The latter study was dedicated to a specific band and is recent (2012). I take this to be a reasonable or even favourable estimate of the expected uncertainty in $CO_2$ spectroscopic line parameters in the OXYCO2 experiment. Some bias correction can be applied by first validating against $CO_2$ measured using techniques accurate to <1 ppmv but the authors would need to discuss this, especially since they insist on using the term 'accuracy' instead of 'precision' in a couple of spots (see above) in the paper.

The authors correctly state (P2L13) that strong correlations exist between retrieved T and retrieved $CO_2$ when retrieving T from $CO_2$ lines and this correlation "prevents" the retrieval of $CO_2$ from these same lines. I believe the authors have the same issue over a large portion of their retrieval range since Figs. 5 and 6 show that the $CO_2$ VMR precision is not changed much if the O2 lines are used or not outside of ∼20-35 km. The 20-35 km corresponds to the region where the information load (IL) is large, whereas outside this altitude range, both above and below, there is a sharp decrease in IL. I believe that the pT information is coming predominantly from $CO_2$ lines (whose IL tends to be larger in the upper stratosphere) and that strong correlations will result. I consider only the region between 20-35 km to be appropriate for retrieval and I believe the authors should "prevent" themselves from retrieving outside of this range, given their retrieval setup. The authors also admit (P8L2) that when they tried the sequential estimation, they could not retrieve $CO_2$ VMR precisions that approach the target value because of problems retrieving T exclusively from the O2 lines. I believe the authors should restrict the vertical range to ∼20-35 km and demonstrate that they can retrieve $CO_2$ in a sequential setup over this 'sweet-spot' range. By going to simultaneous retrievals of $CO_2$ VMR and pT, the authors could be confusing themselves in terms of the benefit of the O2 lines.

Figures 5, 6, and 8 are of low quality (and I am not very picky).

[Figure]

Figure 1 does not serve the intended purpose. It shows me that the O2 lines are not prominent, which contradicts the claim by the authors (P1L26).

A nice addition to Figure 2 would be the IL for TIR + FIR.

I'm impressed by the ability of the authors to calculate horizontal resolutions and by the two sets of IL calculations provided in Fig. 2.

Specific scientific comments

P1L21 The authors should not judge whether the retrieval accuracy is dominated by systematic or random components until they have added major sources of error. Ironically, when some of the same authors including the lead author, presented CO2 error budget for MIPAS (http://www2.fci.unibo.it/~enzop/FILES/CO2.pdf), they included many additional sources of error. 'SPECDB' ironically appears to be spectroscopic database errors, which are important and lead to 1-6% errors according to the online presentation. Other sources of error include 'gain', which is up to 4%. Perhaps, the OXYCO2 instrument will have smaller uncertainty related to gain, but to disregard this source of error entirely seems biased toward minimizing retrieval inaccuracies. Other sources such as 'SPREAD', 'SHIFT', and 'CTMERR' should be considered here if relevant. Something should be said about the impact of thin clouds, particularly on the TIR radiances, since the authors talk about retrieving in the troposphere many times. Wavelength calibration can be a slight problem for FT spectrometers and should be considered. Imperfect knowledge of the instrumental line shape is another source of error that could be considered.

P2L15 The authors could refer to Emmert et al. in Nature Geosci. for the mesospheric ACE-FTS CO2 measurements.

P2L19 There is no such discussion in Bernath et al. (2005).

P4L8 1.5 cm2 sr is a very large throughput. I'm wondering if this is a typo. Could the authors specify the solid angle subtended by the field of view?

P7L1 State clearly whether A and phi are variable or constant along the OC. I assume they are variable from this line.

P9L17 Are these the B values used in Figure 2?

P9L19 values -> absolute values

P24 As mentioned above, there does not appear to be much relaxing of the strong correlation (P2L23) outside of 20-35 km since the red and green lines are not very different. The authors need to show that the strong correlation of T and CO2 VMR is not a problem outside this vertical range.

Minor comments

P1L12 Vibro-rotational -> rovibrational (correct this throughout the paper).

P1L15 operational limb sounders -> an operational limb sounder. (I see MIPAS as the only operational limb sounder).

P1L27 biosphere -> atmosphere

P1L28 on board of -> on board

P1L30 have demonstrated -> "have been demonstrated"

P2L3 Chedein -> Chedin

P2L8 "(e.g." -> "such as" (to avoid too many parentheses).

P2L9 Fischer et al., 2008 -> (Fischer et al., 2008) (repeat for Gille et al.)

P2L12 "...known; assumption..." -> "...known. This assumption..."

P2L15 All leading prepositional phrases should be followed by a comma, e.g. "In these measurements,"

P2L19 ", can be found" -> "can be found"

P2L25 "line strength is" -> "line strengths are"

P2L29 "capable to"->"capable of"

P3L14 No need to cite Fischer et al. again.

P3L16 Hydroxil -> Hydroxyl

P4L9 Can the authors be clearer that this is one component of the overall noise, related to the detector? Also, I understand the units, although they don't appear to be power units and this may confuse some readers.

P4L11 change from square brackets to curvy ones here and in the next line.

P4L11 active -> an active

P4L26 "adapting to our specific needs the algorithm ...2002." -> "adapting the algorithm ... (2002) to our specific needs".

P5L1 increase -> "an increase"

P5L2 "that correspond" -> ", which corresponds"

P5L21 "all the" -> "all of the" (occurs elsewhere).

P5L22 "Fig.s" -> "Figs."

P6L16 round -> root (?)

P6L19 point -> step

P8L33 VCM: spell out acronym here.

P15L2 that -> whom

P15L6 P. F. -> P.-F.

P15L7 Mc-Connel -> McConnell

P15L9 Semeniuxk -> Semeniuk

P15L10 Zou, T. -> Zou, J. (plus add spaces before 'Walkty', 'Wardle', and 'Wehrle')

P15L12 add "and" before last name of final author. (elsewhere as well).

P15L18 Borrowsandi -> Burrows

P15L18 Krner -> Körner

P15L19 SCHIAMACHY -> SCIAMACHY

P17L2 SPIE -> Proc. SPIE 8176

P18 It would be helpful if Table 1 referred to section 2.1 for vertical sampling. A footnote could be used.

Reference

S. A. Tashkun, V. I. Perevalov, R. R. Gamache, J. Lamouroux, CDSD-296, high resolution carbon dioxide spectroscopic databank: Version for atmospheric applications, Journal of Quantitative Spectroscopy & Radiative Transfer 152, 45–73, 2015.

---

## Referee Comment (RC2) · Anonymous Referee #1 · 1 Sep 2016

**Review of "A strategy for the measurement of the CO2 distribution in the stratosphere"**

**General:**

The use of combined FIR and TIR measurements to determine the vertical distribution in the stratosphere is a very interesting idea. The authors have pointed out that the measurement capabilities are available, as is the retrieval software. They have shown that it should be possible, based on measurements in a single orbit, to determine the volume mixing ratio (VMR) from the upper troposphere to 40 or 50 km altitude. The assumptions appear reasonable.

The description of the inadequacy of the sequential retrieval is good.

**Specific points:**

**Introduction:** What is the need and use for such measurements? What are the requirements for determining the accuracy of the stratospheric distribution of $CO_2$? Is an accuracy of 1 ppmv a useful constraint?

P. 4:  line 10:  Please give the low temperatures that are required.
  l. 18:  Indicate that these are the $O_3$ $\nu_2$ band transitions
  l. 20:  are these transitions rotational, continuum, or both?
  l. 25ff- Does this procedure for determining MW's lead to a unique result?  Do the results depend on the order of the seeds chosen?

P. 5: ll. s11ff:  How much information was lost by reducing the number of MWs?  How would the later results have been different if these MW's were included?

P.6: l. 5:  Could the horizontal gradients be treated just as well by using a shorter orbital segment, and moving the segment around the orbit?  Would this save computer resources?
  l. 6:  Clarify that target here refers to the different gases.
  l. 24:  Please say something more about the 2-D averaging kernel- how wide is it?
   A plot or reference would be nice.
  l. 25:  Have you tried doing 1-D retrievals to get the first guess field, then go to 2-D as a correction, or refinement?

P.7: ll. 5,7   These could be stated more clearly by "For each perturbation profile a random value of A is assigned",  and " For each perturbation profile a random value of $\phi$ between 0 and $2\pi$ is assigned"

P. 8: l. 19:  Apparently 401 limb scans are included in a "full orbit".  If overlap to the next orbit is done, it should be stated and if necessary described.
  l. 31:  Spell out VCM first mentioned here (and refer to appendix)
  l. 31:  B values seem very large, especially 80% for $CO_2$.  Are there any model results on the variations of $CO_2$ in the stratosphere?

P. 9:    l. 14:  Discussion of Fig. 4- what is the reason for the vertical pattern of larger differences?   Does this undercut the ability to get a geographic pattern of differences?

        l. 20:  Figs. 5 & 6 need standard deviations as well as mean values.  My understanding of Figs. 5 & 6 is that for B=2 the perturbation is ~0.65%, or about 2.6 ppmv, so that the retrieval has reduced the uncertainty to ~ 1ppmv- is that right?
        l. 32:  The green lines are very interesting, in that they could be implemented by a much simpler instrument than OXYCO2.  How much could the bulge around 30 km be reduced by averaging more orbits?  Why is the bulge smaller in Fig. 6 for B=2?

Again, if a shorter segment of the orbit were used, could more MW's and more spectral points be used, and would this allow better retrievals of $CO_2$?  Would this improve results with only TIR channels?

P. 10    l. 25ff:  If the ozone interference even with OXYCO2 high resolution leads to a systematic error of  ~ 1 ppmv, what is the plan for dealing with this?

P. 12    ll. 8-11:  This is unclear; it seems to say that at the end of an orbit part of the next orbit is added to allow the same views of all scans.  If this is right, please say more clearly.

**Suggestions for Changes in Wording**

There are a number of places where I have suggested alternate wording for smoother English.

P 1,    l. 29  …platforms have been demonstrated

P2,    l. 1:  transitions are clearly visible
        l. 8: Dynamics
        l. 11:  this assumption
        l.12 …features from being used
        l. 23:  suggest "uncouple" in place of 'relax", also "connection" in place of "correlation"
        l. 28:  …suggest using an instrument capable of measuring simultaneously…

P.3,    l.15  Hydroxyl
        l. 23: ENVIronmental
        l. 25: …enables the use of the GMTR…
        ll. 28-29:  …IRLS spectrometer that was designed to …

P, 4,    l. 5:  …recording time, which defines…
         l. 11:  …systems such as Joule-Thompson…
         l. 16:  I would suggest "goal" rather than "target"
         l. 33:  …does not yield an increase …

P. 5,    l. 1:  corresponds
         l. 28:  …highlights how much larger …

P. 6.    l. 14:  root-mean-square

P. 8,    l.7:  Again, I suggest "uncouple" in place of  "relax"; other possibilities include avoid,
bypass, or decouple.
         l. 11:  …T results are constrained…
         ll.23-24: …twice as wide as the measurement…

P. 9,    l.12:  …$CO_2$ fields were negligible…

P. 10,   l. 30:  …possibility of measuring the $CO_2$…

P. 11,   l.1:  suggest "circumvent" instead of "relax"

P. 12,   l. 17,18: …enables the modeling of horizontal …

---

## Author Comment (AC1) · 6 Oct 2016

**Reply to RC1**

**General comments**

We have grouped by topic the reviewer's comments (reported in italic). Our answers follow each topic. A copy of the revised paper was not requested by the editor at this stage of the reviewing process. Therefore we report within square brackets the page and line numbers [P$x$L$y$] where the AMTD paper will be modified. The modified text is then reported in red.

*This paper presents a reasonable strategy for retrieving the vertical distribution of CO2 in the stratosphere. An alternative strategy would be to use solar occultation instead of thermal emission for these same bands. This could improve the shot noise and the vertical resolution since:*

*1) The sun will always be a stronger source of radiation than the Earth's atmosphere at any wavelength.*

*2) For solar occultation, the vertical resolution is not tied to the vertical structure of the temperature (see Fig. 7 bottom left of Carlotti et al.), and therefore would not worsen severely in the tropopause region as is the case for thermal emission as expected.*

The purpose of this paper is not to compare the performance of emission and occultation experiments with respect to the measurement of atmospheric $CO_2$. Our purpose is to show the potentiality of a new strategy that, to the best of our knowledge, was never investigated before.

*For CO2, given the main interest of the authors to observe the long-term slight increases in VMR in the stratosphere, frequent measurements are not required and thus space-based solar occultation could be applicable.*

We never mention our "main interest to observe the long-term slight increases in VMR in the stratosphere". The "long-term slight increases" could be a constraint to knowledge imposed by the lack of information about stratospheric $CO_2$.

*The authors could consider the O2 lines in the TIR as well, which are useful in the stratosphere up to ~38 km, based on ACE-FTS O2 retrieval accuracy. This could alleviate the need for two detectors regardless of whether thermal emission or solar occultation is used.*

We did not consider the use of the $O_2$ band around 1500 cm$^{-1}$ because, as we point out in sect. 2.1, our concern was to avoid as much as possible the effect of interfering transitions and we expect the spectrum to be more crowded in the region of

rovibrational transitions than in the one of pure rotational transitions. Moreover, the intensity of the 1500 cm$^{-1}$ transitions is 15-20 times lower than that of the FIR transitions. This can be appreciated by comparing the lower panel of Fig. 1 with the figure below that reports a simulation of the 1500 cm$^{-1}$ band for the same observation conditions.

[Figure]

*There are three main problems with this paper:*
*1) The error budget is incomplete, specifically with regard to sources of systematic uncertainty.*
*the authors present their method as one that has small systematic uncertainties and an accuracy of 1 ppm (P2L30 & P12L3), yet ignore many significant systematic sources of uncertainty and consider only minor ones. The major revisions involve accounting for more sources of systematic error. Furthermore, the statement that CO2 VMR is retrievable to 1 ppmv between 10 and 50 km is grossly misleading in my opinion. There is no point to getting the community excited about an instrument that can supposedly measure CO2 profiles to ~1 ppm, when it hinges on line intensities of CO2 to be measured to 0.25%....... The combined biases in CO2 and O2 spectroscopy could either cancel or lead to a 2% bias in the worst case.*
*Some bias correction can be applied by first validating against CO2 measured using techniques accurate to <1 ppmv but the authors would need to discuss this, especially since they insist on using the term 'accuracy' instead of 'precision' in a couple of spots (see above) in the paper.*

Following these reiterated criticisms, we decided to thoroughly review the error sources that were considered in the MWs selection process. The outcome is that, for

an academic study, the choice was to assess the performance of an ideal instrument so that no instrumental errors were considered. About the spectroscopic data uncertainties, they were also neglected on the basis of the assumption that they can be measured with the desired accuracy. In order to account for the above considerations Sect. 2.2 will be modified at [P4L27] by adding after "Dudhia et al., (2002)":

For the purpose a set of error sources must be defined and quantified in order to evaluate the uncertainty associated to each spectral point. Here we have considered errors deriving from the VMR uncertainty of all of the atmospheric constituents, and the error deriving from the Non Local Thermal Equilibrium (NLTE) conditions when they are not modeled in the retrieval system. Instrumental and spectroscopic errors have been omitted in this academic study by assuming that, in the case of operational implementation, they will have to be assessed on the basis of the existing technology. According to these statements we will introduce the following changes:

In the Abstract "accuracy" will be modified into precision at [P1L17], and the period starting at [P1L21] will be:

The error budget, estimated for the test-case of an ideal instrument and neglecting the spectroscopy errors, indicates that, in the 10-50 km altitude range, the total error of the $CO_2$ fields is set by the random component. This is also the case at higher altitudes provided….

In the Introduction section at [P2L30] "target accuracy" will be "target precision".

In Sect. 4.4 the period starting at [P10L17] becomes:

This budget indicates that, among the considered error sources (see Sect. 2.2), the dominant components…..

In the "Conclusions" section the paragraph starting at [P11L29] will be:

The assessment of the systematic errors considered in this study (VMR of the atmospheric constituents and NLTE conditions) shows that below 50 km their contribution to the total error…..

The sentence starting at [P12L2] will be omitted in the revised paper.

*The claim that dedicated spectroscopic measurements will be made in the future is not acceptable to me for the present manuscript. I would assume the spectroscopists previously involved in measuring line intensities were dedicated to achieving the best accuracies possible. See table 3 of Tashkun et al. (2015). Searching through the systematic uncertainty column (2nd last column) of this table, I see values as low as 2% (e.g. by Delière et al., 2012) for the lines in a region overlapping the spectral region proposed by the authors. The latter study was dedicated to a specific band and is recent (2012). I take this to be a reasonable or even favourable estimate of the expected uncertainty in CO2 spectroscopic line parameters in the OXYCO2 experiment.*

To date, $CO_2$ line intensities (as well as other spectroscopic parameters) can be determined with uncertainties that are better than 0.3 % (see e.g. Oleg L. Polyansky et al., Phys. Rev. Lett. 114, 243001 (2015), G. Casa et al., J. Chem. Phys. 130, 184306 (2009)). Apart from this consideration, our paper should not be read as a proposal to a space agency. As stated above we investigate the potentiality of a new strategy that, if ever considered for operational implementation, will have to take into account the state of the art of the existing technology for both instrumental errors (see previous answer) and the laboratory measurements. We will remark this point in the "Introduction" section by adding after the period at [P3L6]:

This academic study is directed to assess the intrinsic capability of the proposed approach irrespective of some technological aspects that need to be evaluated when an operational experiment is considered.

On the other hand, the reviewer seems to neglect that, as specified at [P8L9-11], "the dominant information about T comes from the shape of the Planck function rather than from the dependence of the line strengths from T" (see Carlotti et al., 2013 where this statement is better quantified).

*Also, it hinges on O2 line intensities to be measured extremely accurately: a 1% bias in O2 line intensities will lead to a 70 m bias in TH (or a 1% bias in pressure). This will translate to a ~1% bias in CO2, much larger than the sources of systematic error that the authors have selected.*

We retrieve pressure (P) together with temperature (T) and the VMR targets. Since we do not assume hydrostatic equilibrium, any altitude bias translates into a bias on the retrieved P profiles. As reported in Sect. 4.1 the errors on P have a negligible impact on the $CO_2$ VMRs. This consideration already appears in Sect. 4.4.

*O2 does not appear to be adding much p T information outside of the 20-35 km range, raising the question about strong correlations between CO2 VMR and T.*
*The authors correctly state (P2L13) that strong correlations exist between retrieved T and retrieved CO2 when retrieving T from CO2 lines and this correlation "prevents" the retrieval of CO2 from these same lines. I believe the authors have the same issue over a large portion of their retrieval range since Figs. 5 and 6 show that the CO2 VMR precision is not changed much if the O2 lines are used or not outside of ~20-35 km.*

The reviewer refers to [P2L13] in the "Introduction" section where we pose the problem. We carried out the test with and without $O_2$ transitions just "to assess whether and to what extent the FIR observations are necessary and contribute to the precision obtained in our retrieval tests" [P9L27-28]). In our opinion this was a

question to answer and Figs. 5 and 6 provide the answer. These figures are self-explaining and we don't think further comments are necessary in the revised text.

*I consider only the region between 20-35 km to be appropriate for retrieval and I believe the authors should "prevent" themselves from retrieving outside of this range, given their retrieval setup.*

Part of this study is also the assessment of the altitude range where $CO_2$ can be retrieved. In order to do this, the vertical region of the retrieval had to be as wide as possible. The reviewer writes the above statement only after having inspected Figs. 5 and 6 that are results of our study. Nevertheless, these figures show that the altitudes outside the 20-35 km range could be appropriate for the retrieval of $CO_2$ VMRs even without the FIR contribution.

*The 20-35 km corresponds to the region where the information load (IL) is large, whereas outside this altitude range, both above and below, there is a sharp decrease in IL. I believe that the pT information is coming predominantly from CO2 lines (whose IL tends to be larger in the upper stratosphere) and that strong correlations will result.*

This is a reasonable interpretation. On the other hand the IL analysis provides indications about the distribution of information with respect to a single retrieval target. Therefore we prefer not to include this conjecture in the text because, as we notice at [P10L10-11]; "the complex interdependence between the many variables of the MTR inversion makes difficult the interpretation" of our results in terms of IL.

*The major revisions involve accounting switching back to sequential retrievals.*
*The authors also admit (P8L2) that when they tried the sequential estimation, they could not retrieve CO2 VMR precisions that approach the target value because of problems retrieving T exclusively from the O2 lines. I believe the authors should demonstrate that they can retrieve CO2 in a sequential setup over this 'sweet-spot' range. By going to simultaneous retrievals of CO2 VMR and pT, the authors could be confusing themselves in terms of the benefit of the O2 lines.*

In our opinion the simultaneous retrieval of $CO_2$ VMR and pT (and also $H_2O$ VMR) is not a source of confusion but a powerful tool widely used in the analysis of remote sensing measurements. In our study we have verified that the sequential estimation is not suitable to get the required precision (even in the sweet-spot) because of the insufficient precision of T provided by only the $O_2$ lines. We have demonstrated that the problem can be overcome by exploiting at best the available retrieval techniques such as the 2-D approach (that merges information from adjacent limb-scans and is not applicable to sun occultation measurements since they do not observe along the

orbit plane) and the MTR approach that merges information about pT from the spectral features of different atmospheric constituents ($O_2$, $CO_2$ and $H_2O$ in our case). Moreover, the benefit of using the $O_2$ within a Multi-Target, instead of a sequential Retrieval, is not only the independence of its VMR from the $CO_2$ VMR but also (as specified at the end of Sect. 4.1) the advantage obtained by joining FIR and TIR observations that makes the sampling of the Plank function more extensive and nails down the temperature more efficiently than using TIR or FIR alone.

Therefore we don't see any reason to switch to a strategy that we know to be unsuited to our objective.

*Figures 5, 6, and 8 are of low quality (and I am not very picky).*

We have generated a new version of these figures (of better quality) for the revised manuscript.

*Figure 1 does not serve the intended purpose. It shows me that the O2 lines are not prominent, which contradicts the claim by the authors (P1L26).*

The reviewer is right (we assume he refers to P2L26). The lower panel of Fig. 1 is meant to show an overall picture of the atmospheric pure rotational spectrum of $O_2$ that we could not find in the literature. The statement about prominence of $O_2$ transitions is taken from the reference "Carli and Carlotti, 1992" that we will add in the revised manuscript after the period at [P2L26] and in the "References" section. In the upper panel of the figure we do not see the prominent lines of $O_2$ because of both the insufficient frequency scale expansion and the low TH. After the period at [P2L29] we will add in the revised paper:

The compressed scale of Fig. 1 prevents the identification of $O_2$ lines in the upper panel. However the comparison of the two panels shows that, below 170 cm$^{-1}$, the intensity of the $O_2$ lines matches the maximum emission of the atmospheric spectrum. On the basis of these considerations…..

The upper panel of Fig.1 is also meant to show the "steep growth of the Planck function" that we mention at [P3L1].

*A nice addition to Figure 2 would be the IL for TIR + FIR.*

We considered this further panel but we decided not to include it in Fig. 2 because, due to the higher values of the IL in the TIR, the quadratic summation combination-law (see the matrix algebra in appendix A) and the resolution of the color palette makes the TIR+FIR map indistinguishable from the upper right panel. After the period at [P5L30] in the revised manuscript we will add:

(The IL with respect to the T of combined set of 15 MWs is not shown because, due to the different magnitudes, the quadratic-summation combination law makes this map quite similar to the upper right panel).

**Specific scientific comments**

*P1L21 ..... Something should be said about the impact of thin clouds, particularly on the TIR radiances, since the authors talk about retrieving in the troposphere many times.....*

The issue of clouds is not considered as it is a known problem for TIR radiances in general. No specific effect is expected on OXYCO2.
The answers and the modifications proposed within the above "general comments" cover the remainder of this reviewer's comment (not reported).

*P2L15 The authors could refer to Emmert et al. in Nature Geosci. for the mesospheric ACE-FTS CO2 measurements.*

The suggested reference will be added at [P2L15] and in the "References" section.

*P2L19 There is no such discussion in Bernath et al. (2005).*

True. We will correct the reference that is to Foucher et al., 2011.

*P4L8 1.5 cm2 sr is a very large throughput. I'm wondering if this is a typo. Could the authors specify the solid angle subtended by the field of view?*

This was a typo. The right number is 0.015 $cm^2$ sr. The whole period will be reassembled after the period at [P4L7] as:
The NESR requirement assumed for this study can be obtained with a detector-noise limited spectrometer with an optical layout, similar to the one used in SAFIRE, with an optical throughput of 0.015 $cm^2$ sr, and using 4.2 K cooled detectors.
This modification also accounts for the reviewer's request that we report below within the "minor comments".

*P7L1 State clearly whether A and phi are variable or constant along the OC. I assume they are variable from this line.*

The required statement will be introduced by modifying the [P7L9] as:

For each perturbation profile along the orbit a random value of $A \leq B$ and a random value of $\phi$ between 0 and $2\pi$ is assigned.

*P9L17 Are these the B values used in Figure 2?*

Figure 2 shows IL maps that, as specified at P5L20-21, have been calculated using the reference climatological atmosphere. There is no reason to perturb the reference atmosphere for the IL calculations.

*P9L19 values -> absolute values*

Done

*P24 As mentioned above, there does not appear to be much relaxing of the strong correlation (P2L23) outside of 20-35 km since the red and green lines are not very different. The authors need to show that the strong correlation of T and CO2 VMR is not a problem outside this vertical range.*

See the answers about this point provided within the "general comments".

**Minor comments**

*P1L15 operational limb sounders -> an operational limb sounder. (I see MIPAS as the only operational limb sounder).*

MIPAS is the only operational space-borne TIR limb sounder. Other operational limb sounders exist in different spectral regions (e.g. MLS) and on different platforms (balloons or aircrafts). In our paper we exploit the heritage of SAFIRE which is a FIR aircraft-borne limb sounder.

*P1L27 biosphere -> atmosphere*

$CO_2$ affects the radiative budget of the entire "zone of life on Earth" (the biosphere).

*P4L9 Can the authors be clearer that this is one component of the overall noise, related to the detector? Also, I understand the units, although they don't appear to be power units and this may confuse some readers.*

For this feasibility study we have considered a detector-noise limited spectrometer (which is consistent with the FTS choice) having the main requirement of 5nW for the spectroscopic measurement. We will modify the sentence starting after the period at [P4L7] as:

The NESR requirement assumed for this study can be obtained with a detector-noise limited spectrometer with an optical layout, similar to the one used in SAFIRE, with an optical throughput of 0.015 $cm^2$ sr, and using 4.2 K cooled detectors.

All of the minor comments that we don't mention will be implemented in the revised text as suggested by the reviewer.

---

## Author Comment (AC2) · 6 Oct 2016

**Reply to RC2**

The reviewer's comments are reported in italic; our answers follow each comment. A copy of the revised paper was not requested by the editor at this stage of the reviewing process. Therefore we report within square brackets the page and line numbers [P$x$L$y$] where the AMTD text will be modified. The modified text is then reported in red.

**Specific points**

***Introduction***:
*What is the need and use for such measurements?*

The role and the importance of atmospheric $CO_2$ are well established concepts. We concisely recall them in the first lines of the "introduction" section and we cite a literature (in the lines that follow) where the argument is discussed. We consider unnecessary to include further details in our text.

*What are the requirements for determining the accuracy of the stratospheric distribution of CO2? Is an accuracy of 1 ppmv a useful constraint?*

Please note that, in the revised paper, 1 ppmv will be indicated as the target precision of the retrieved $CO_2$ VMRs. The reference adopted to set this constraint is the rate of $CO_2$ VMR increase in the troposphere that is ~ 2.5 ppmv per year. On the other hand, considering the present lack of measurements of stratospheric $CO_2$ distributions, 1 ppmv would represent a major step.

*P. 4*
*line 10: Please give the low temperatures that are required.*

In the revised paper the period starting at [P4L7] will be reassembled as:
The NESR requirement assumed for this study can be obtained with a detector-noise limited spectrometer with an optical layout, similar to the one used in SAFIRE, with an optical throughput of 0.015 cm$^2$ sr, and using 4.2 K cooled detectors.

*l. 18: Indicate that these are the O3 v2 band transitions*
*l. 20: are these transitions rotational, continuum, or both?*

In the revised paper we will specify: "the main interference to the $CO_2$ spectral features is due to the $O_3$ $v_2$ band rovibrational transitions"

*l. 25ff- Does this procedure for determining MW's lead to a unique result? Do the results depend on the order of the seeds chosen?*

The algorithm for the definition of seeds is based on the derivative of spectral points with respect to target parameters (Jacobian matrix) within the considered spectral interval. The derivative is a measure of the information content that provides an objective criterion which is not driven by the user. Therefore the order of the seeds (and the result) is unique for this procedure.

*P. 5 ll. s11ff: How much information was lost by reducing the number of MWs? How would the later results have been different if these MW's were included?*

We did not carry out the test with all the MWs generated by the selection algorithm. However, we verified that the inclusion of two additional MWs led to minor changes in the results. This behavior is expected if the previous selection has been operated by choosing the highest IL values that cover at best the altitude range. In this case the inclusion of further MWs corresponds to introduce further terms in a quadratic summation (see the matrix algebra in appendix A) where the largest terms, that dominate, are already present.

*P.6*
*l. 5: Could the horizontal gradients be treated just as well by using a shorter orbital segment, and moving the segment around the orbit? Would this save computer resources?*

The strategy to break the 2-D retrieval of the full orbit into a set of retrievals over segments moving around the orbit requires less computer memory but probably not less CPU time. Actually, the orbit segments cannot be adjacent but overlapping because the results close to the edges of an orbit segment must be discarded. This implies that a number of limb-scans must be processed twice. On the other hand, the problems linked to the computer resources can be overcome with an ad-hoc matrix compression algorithm and the corresponding matrix algebra (see Carlotti et al., 2001a, Carlotti et al., 2006).

*l. 6: Clarify that target here refers to the different gases.*

We did not catch the point. Any geophysical parameter can be a target. In our case we have two gases ($CO_2$ and $H_2O$), pressure, temperature and atmospheric continuum at the frequency of the analyzed MWs. The state vector is specified in the first two lines of Sect 4.2.

*l. 24: Please say something more about the 2-D averaging kernel- how wide is it? A plot or reference would be nice.*

The 2-D averaging kernel is a square matrix whose dimension is the number of retrieval parameters that, as specified at [P8L29], is 24840. A map of such a huge matrix does not provide useful information especially if it refers to a MTR where several targets are merged together. More meaningful are the maps of the spatial resolution for specific targets as those reported in Fig. 7. The reference for this subject is Carlotti et al., (2007).

*l. 25: Have you tried doing 1-D retrievals to get the first guess field, then go to 2-D as a correction, or refinement?*

Not in this study. However our experience indicates that the additional 1-D step could lead to save one (or two) 2-D retrieval iteration but does not improve the results.

*P.7: ll. 5,7 These could be stated more clearly by "For each perturbation profile a random value of A is assigned", and " For each perturbation profile a random value of φ between 0 and 2π is assigned"*

In the revised paper we will reassemble the statement at [P7L9] as indicated by the reviewer.

*P. 8*

*l. 19: Apparently 401 limb scans are included in a "full orbit". If overlap to the next orbit is done, it should be stated and if necessary described.*

Please note that, in Table 1, the horizontal sampling of ~110 km was a typo. We have now corrected to 100 km this value.
In our orbit the average OC of the first limb scan is about 0.24 deg. The separation between limb scans is about 0.9 deg. The limb scan 401 has OC of about 359.26 deg. So there is no overlap.

*l. 31: Spell out VCM first mentioned here (and refer to appendix)*

Done.

*l. 31: B values seem very large, especially 80% for CO2. Are there any model results on the variations of CO2 in the stratosphere?*

The *B* values have not been chosen on the basis of climatological or model variability but with the purpose to test the robustness of the retrieval system even in "extreme" conditions. This concept is already expressed at [P9L7]. As specified at [P5L21], we use the model described in Remedios et al., (2007).

*P. 9*

*l. 14: Discussion of Fig. 4- what is the reason for the vertical pattern of larger differences? Does this undercut the ability to get a geographic pattern of differences?*

In Fig. 4 the largest differences occur in the troposphere (that can be identified by looking at the temperature distribution in the lower-right map of Fig. 2) and are explained by the known opacity of this region at the considered wavelengths. The vertical pattern of the differences represents the statistical fluctuation deriving from the 20 random perturbations applied to the initial guess profiles at each OC. We hope to have correctly interpreted the reviewer's point.

*l. 20: Figs. 5 & 6 need standard deviations as well as mean values.*

We have calculated the standard deviation and the uncertainty for each of the curves reported in Figs. 5 and 6 (the figures below show the % uncertainties as a function of altitude). However, including further curves sensibly worsen the clarity of Figs. 5 and 6 so that we prefer not to show them. In the revised paper, after the period at [P10L2], we will add:
The uncertainty of the average differences plotted in Figs. 5 and 6 ranges between 1.2% and 2.3%.

[Figure]

Fig. 5 red lines

Fig. 5 green lines

Fig. 6 red lines

Fig. 6 green lines

*My understanding of Figs. 5 & 6 is that for B=2 the perturbation is ~0.65%, or about 2.6 ppmv, so that the retrieval has reduced the uncertainty to ~ 1ppmv- is that right?*

Yes. However it is not fully correct to refer to 2.6 ppmv as an "uncertainty"; it is the average error made when assuming the initial guess as "true" value.

*l. 32: The green lines are very interesting, in that they could be implemented by a much simpler instrument than OXYCO2.*

Yes. It depends on the importance attributed to the factor of about two around 30 km.

*How much could the bulge around 30 km be reduced by averaging more orbits?*

We focused the paper on the performance of individual retrieved profiles. We omitted to discuss the issue of averaging strategies that, of course, can be used to increase the precision. Averaging can be done over more orbits (taking care of averaging profiles with common geo-location); in this case there is a loss in time resolution. Averaging can also be implemented within the single orbit over geographical regions; in this case the loss is in space resolution.

*Why is the bulge smaller in Fig. 6 for B=2?*

By considering the complexity of the retrieval scenario we don't have a unique explanation for this statistical behavior.

*Again, if a shorter segment of the orbit were used, could more MW's and more spectral points be used, and would this allow better retrievals of CO2?*
*Would this improve results with only TIR channels?*

The answer provided above (relative to the question for P5) is valid also for orbit segments.

*P. 10 l. 25ff: If the ozone interference even with OXYCO2 high resolution leads to a systematic error of ~ 1 ppmv, what is the plan for dealing with this?*

With the exception of ACE measurements, the present knowledge of stratospheric $CO_2$ distribution comes from model calculations. On the light of this we consider the precision of 1 ppmv a result that, if achieved by an operational experiment, would represent a major improvement. The impact of ozone interference could be reduced by including its VMR in the state vector (not tested) and with a better knowledge of ozone climatology (that can be exploited to reduce its a-priori error in the optimal-estimation process).

*P. 12 ll. 8-11: This is unclear; it seems to say that at the end of an orbit part of the next orbit is added to allow the same views of all scans. If this is right, please say more clearly.*

(We identified the part of the paper this comment refers to even if the reviewer seems to have indicated the wrong lines of the AMTD text). In the geo-fit analysis a full orbit of measurements is built by starting from the limb-scan whose tangent heights (TH) directly follows the North (or South) Pole having OC=0. Limb-scans are then added until the OC of its THs does not overpass OC=360. At this point the loop of overlap between nearby sequences closes with the atmospheric parcel sounded by the first limb-scans observed again by the last limb-scans even if after the orbit period (101 min) and from a different point of view (angular difference of about 20 deg).

All but one of the **suggestions for changes in wording** will be implemented in the revised text. The exception is the use of "connection" instead of "correlation" at [P2L23]. The reason is that "correlation" has a specific meaning in the mathematics of the inversion algorithm.